# HypCL: Adapting CLIP in Hyperbolic Space for Continual Learning

**Quan Cheng** [1 2] **Hao Yu** [1 2] **Da-Wei Zhou** [1 2] **Lijun Zhang** [1 2]

## Abstract

Recently, vision-language models (e.g., CLIP) have been increasingly adopted for continual learning to mitigate catastrophic forgetting. However, existing CLIP-based methods typically freeze the backbone to preserve pre-trained knowledge, which limits the model's ability to learn discriminative features for downstream tasks. In this paper, we introduce HypCL, a parameter-efficient framework that continually adapts CLIP in hyperbolic space for continual learning. Our key insight is that the exponentially expanding capacity of hyperbolic geometry naturally accommodates the growing class space and promotes stronger inter-class separation. Specifically, HypCL attaches task-specific adapters and composes their updates sequentially in the Poincaré ball. To exploit the enhanced feature separability of hyperbolic geometry, HypCL maintains visual prototypes computed from the adapted features, which serve as stable anchors for calibrating predictions at inference. Extensive experiments on standard class-incremental benchmarks demonstrate that HypCL consistently outperforms existing CLIP-based continual learning methods.

## 1. Introduction

Deep neural networks have achieved remarkable success under the assumption that data is independent and identically distributed (i.i.d.) (LeCun et al., 2015; Goodfellow et al., 2016). However, real-world applications often violate this premise and present data in sequential streams with shifting distributions. Continual Learning (CL) seeks to address this by enabling models to accumulate knowledge over time (Wang et al., 2024; Zhou et al., 2024b;c). The central challenge in CL is the stability-plasticity dilemma. The model must be plastic enough to learn new tasks yet stable enough to prevent catastrophic forgetting of previous knowledge (McCloskey & Cohen, 1989; Kirkpatrick et al., 2017).

In recent years, Vision-Language Models (VLMs) such as CLIP (Radford et al., 2021) have demonstrated remarkable capabilities. CLIP learns a joint embedding space where visual and textual representations are aligned. This enables strong zero-shot generalization to novel categories. Moreover, the text embeddings encode rich semantic relationships, including implicit hierarchical structures among concepts. These properties make CLIP a promising foundation for continual learning (Huang et al., 2025). Existing CLIP-based CL methods typically freeze the backbone to preserve pre-trained knowledge while training lightweight modules such as prompts for task adaptation (Huang et al., 2024; Yu et al., 2025; Zhou et al., 2025a). However, this conservative strategy limits the model's ability to learn discriminative features for specialized categories. Full fine-tuning, on the other hand, causes severe representation drift that destroys the learned alignment.

Beyond the adaptation strategy, the geometry of the embedding space also contributes to this difficulty. Standard methods operate in Euclidean space, where volume grows polynomially with radius. As the number of classes increases across tasks, the feature space becomes progressively crowded, making it difficult to maintain separation between old and new classes. Hyperbolic space offers a principled alternative. Its exponential volume growth provides naturally expansive capacity to accommodate new classes without compressing prior representations (Nickel & Kiela, 2017; Hamann, 2018; Desai et al., 2023). Moreover, its constant negative curvature induces a strong inductive bias for hierarchical organization, which complements the semantic structure inherent in CLIP's text embeddings (Peng et al., 2025b). By adapting CLIP in hyperbolic space, we can leverage this cross-modal semantic hierarchy while enabling flexible adaptation.

Building on these insights, we introduce HypCL, a parameter-efficient framework for continually adapting CLIP in hyperbolic space. HypCL freezes the pre-trained CLIP encoders and attaches lightweight adapters to the attention weights of the visual encoder. Each task learns an

[1] State Key Laboratory of Novel Software Technology, Nanjing University, Nanjing, China [2] School of Artificial Intelligence, Nanjing University, Nanjing, China. Correspondence to: Lijun Zhang <zhanglj@lamda.nju.edu.cn>.

*Proceedings of the 43rd International Conference on Machine Learning*, Seoul, South Korea. PMLR 306, 2026. Copyright 2026 by the author(s).

independent update, which is frozen after training and composed with previous updates through Möbius addition in the Poincaré ball. HypCL further aligns the classifier with this geometry by using hyperbolic distances between adapted image features and frozen text embeddings, and refines the final prediction with visual-prototype logits. Together, these components enable HypCL to learn more separable representations across tasks, as we empirically verify in our experiments.

We summarize our contributions as follows.

- We propose HypCL, a parameter-efficient framework that enables continual adaptation of CLIP in hyperbolic space through task-specific updates composed by Möbius operations.
- We introduce a visual-prototype calibration mechanism that exploits the enhanced feature separability of hyperbolic representations to refine prediction logits during inference.
- We conduct extensive experiments on standard class-incremental benchmarks, demonstrating that HypCL consistently outperforms state-of-the-art CLIP-based continual learning methods.

## 2. Related Work

In this section, we review related work on continual learning and hyperbolic learning.

### 2.1. Continual Learning

Continual learning (CL) studies how to learn from sequential tasks while retaining previously acquired knowledge (Wang et al., 2024; Zhou et al., 2024c). Traditional CL methods typically initialize models with random weights and train from scratch, falling into several categories. Regularization-based approaches constrain updates to parameters important for past tasks (Kirkpatrick et al., 2017; Zenke et al., 2017) or use knowledge distillation to preserve old behaviors (Li & Hoiem, 2017). Replay-based approaches rehearse stored exemplars or generated samples to reduce forgetting (Rebuffi et al., 2017; Chaudhry et al., 2019a;b), but often introduce memory and privacy concerns. Architecture-based methods allocate task-specific capacity to alleviate interference (Yoon et al., 2018; Li et al., 2019; Liang & Li, 2023). Optimization-based methods project gradients to avoid interference with previous tasks (Farajtabar et al., 2020; Saha et al., 2021; Wang et al., 2021).

With the rise of large pre-trained models, CL has shifted from training from scratch to adapting foundation representations (Zhou et al., 2024b). Early efforts in this direction leverage pre-trained Vision Transformers (ViTs) (Dosovitskiy, 2020). A common strategy freezes most backbone parameters and learns lightweight modules to reduce in-terference (McDonnell et al., 2023; Zhang et al., 2023). Prompt-based methods select or compose learnable prompts over time (Wang et al., 2022a;b; Smith et al., 2023; Lu et al., 2024). Parameter-efficient fine-tuning mechanisms such as adapters and low-rank updates offer similar benefits (Gao et al., 2023; Liang & Li, 2024; Wu et al., 2025).

More recently, vision-language models such as CLIP (Radford et al., 2021) have been increasingly adopted for continual learning. Many CLIP-based CL methods freeze both encoders to preserve pre-trained alignment and adapt only small components such as prompts (Wang et al., 2023) or lightweight adapters (Huang et al., 2024; Zhou et al., 2025a). Although freezing the backbone improves stability, it restricts the model's ability to learn discriminative features for new classes, especially under long task sequences. This limitation motivates our approach of enabling controlled adaptation of CLIP while maintaining stability.

### 2.2. Hyperbolic Learning

Hyperbolic learning explores representation and optimization in non-Euclidean spaces with constant negative curvature (He et al., 2025). Hyperbolic space can be viewed as a continuous analogue of tree-like structures (Bridson & Haefliger, 2013; Hamann, 2018). Unlike Euclidean geometry, hyperbolic geometry provides exponentially expanding capacity as the radius increases. This makes it well suited for representing hierarchies with low distortion (Desai et al., 2023). In hyperbolic embeddings, the distance to the origin naturally correlates with semantic granularity, allowing concepts to be organized by depth or specificity (Peng et al., 2025b). Among various models of hyperbolic geometry, the Poincaré ball model is widely adopted for its natural ability to capture hierarchical structures (Nickel & Kiela, 2017).

The advantages of hyperbolic space have motivated the adoption of hyperbolic representations in computer vision and multimodal learning. Previous work shows benefits for tasks with implicit hierarchies, including fine-grained classification (Xu et al., 2023), few-shot learning (Khrulkov et al., 2020; Gao et al., 2021), and semantic segmentation (Atigh et al., 2022). In vision-language learning, hyperbolic spaces have been explored to capture cross-modal semantic hierarchies (Desai et al., 2023; Ibrahimi et al., 2024) and enable parameter-efficient adaptation (Peng et al., 2025a). Despite these advances, applying hyperbolic learning to continual learning remains under-explored. Our work addresses this gap by introducing a hyperbolic adaptation mechanism tailored to continual learning with CLIP.

## 3. Preliminaries

This section formalizes the problem setting and reviews the necessary background on CLIP and hyperbolic geometry.

### 3.1. Problem Setting

We consider the Class-Incremental Learning (CIL) setting with vision-language models, such as CLIP (Radford et al., 2021). CIL requires a model to sequentially learn from a stream of tasks while retaining knowledge of previously learned classes. We use task to denote each incremental learning step throughout the paper. Formally, we consider $T$ tasks indexed by $t \in \{1, \ldots, T\}$. At each task $t$, the learner is provided with a labeled dataset $\mathcal{D}_t = \{(\mathbf{x}_i^t, y_i^t)\}_{i=1}^{n_t}$, where $\mathbf{x}_i^t \in \mathcal{X}$ denotes an input sample and $y_i^t \in \mathcal{Y}_t$ is its corresponding label. The label sets across tasks are mutually disjoint, i.e., $\mathcal{Y}_t \cap \mathcal{Y}_{t'} = \varnothing$ for all $t \neq t'$. At evaluation time, the model is tested on all classes observed up to the current task, denoted by $\mathcal{Y}_{\leq t} = \bigcup_{k=1}^{t} \mathcal{Y}_k$, without access to task identities. We focus on the exemplar-free setting, where no historical samples are stored after each task. Consequently, when learning task $t$, the algorithm can only access the current dataset $\mathcal{D}_t$. The objective is to learn a predictor $f_t : \mathcal{X} \rightarrow \mathcal{Y}_{\leq t}$ that minimizes the expected misclassification risk over the cumulative data distribution:

$$f_t^* = \arg\min_{f \in \mathcal{H}} \mathbb{E}_{(\mathbf{x},y) \sim \bigcup_{k=1}^{t} \mathcal{D}_k} \left[ \mathbb{I}\big(y \neq f(\mathbf{x})\big) \right], \quad (1)$$

where $\mathcal{H}$ denotes the hypothesis space and $\mathbb{I}(\cdot)$ is the indicator function. A fundamental challenge in this setting is catastrophic forgetting: updating the model on $\mathcal{D}_t$ can significantly degrade performance on previously learned classes $\mathcal{Y}_{<t}$.

CLIP (Radford et al., 2021) is a vision-language model pre-trained on large-scale image-text pairs via contrastive learning, demonstrating strong zero-shot generalization to novel categories. It consists of an image encoder $g_{\text{img}} : \mathcal{X} \rightarrow \mathbb{R}^d$ and a text encoder $g_{\text{text}} : \mathcal{T} \rightarrow \mathbb{R}^d$, both mapping their respective inputs into a shared $d$-dimensional embedding space:

$$\mathbf{z} = g_{\text{img}}(\mathbf{x}) \in \mathbb{R}^d, \quad \mathbf{w} = g_{\text{text}}(\mathbf{t}) \in \mathbb{R}^d. \quad (2)$$

For classification, CLIP constructs a textual prompt $\mathbf{t}_c$ for each class $c \in \mathcal{Y}_{\leq t}$ using a fixed template (e.g., "a photo of a [CLASS]") and computes the corresponding text embedding $\mathbf{w}_c = g_{\text{text}}(\mathbf{t}_c)$. Given a test image $\mathbf{x}$, the predicted class distribution is obtained via temperature-scaled cosine similarity:

$$p(y = c \mid \mathbf{x}) = \frac{\exp\big(\cos(\mathbf{z}, \mathbf{w}_c)/\tau\big)}{\sum_{j \in \mathcal{Y}_{\leq t}} \exp\big(\cos(\mathbf{z}, \mathbf{w}_j)/\tau\big)}. \quad (3)$$

In CLIP-based continual learning, it is common practice to freeze the text encoder and cache $\{\mathbf{w}_c\}_{c \in \mathcal{Y}_{\leq t}}$ for all observed classes, which preserves the pre-trained language-vision alignment and provides a stable anchor for the expanding class space. Our goal is to adapt CLIP to achieve better discriminability on downstream tasks while preventing catastrophic forgetting of previously learned classes.

### 3.2. Hyperbolic Geometry

Hyperbolic geometry is a non-Euclidean geometry characterized by constant negative curvature. Unlike Euclidean space, where volume grows polynomially with radius, hyperbolic space exhibits exponential volume growth. This property makes hyperbolic space particularly well-suited for embedding hierarchical structures with low distortion, as the available space expands rapidly toward the boundary. In the context of continual learning, this exponentially expanding capacity can naturally accommodate an increasing number of classes without compressing prior representations.

Among various models of hyperbolic geometry, we adopt the Poincaré ball model $(\mathbb{B}_c^n, g^{\mathbb{B}_c})$ with curvature $-c$ $(c > 0)$ due to its computational convenience and differentiability (Hu et al., 2024; Ramasinghe et al., 2024; Peng et al., 2025b). The Poincaré ball is defined as:

$$\mathbb{B}_c^n := \{\mathbf{x} \in \mathbb{R}^n : c\|\mathbf{x}\|^2 < 1\}, \quad (4)$$

equipped with the Riemannian metric $g^{\mathbb{B}_c} = \lambda_{c,\mathbf{x}}^2 g^E$, where $\lambda_{c,\mathbf{x}} = \frac{2}{1-c\|\mathbf{x}\|^2}$ is the conformal factor and $g^E = \mathbf{I}_n$ denotes the Euclidean metric, and $\|\cdot\|$ denotes the Euclidean norm.

Since standard Euclidean operations (e.g., addition and scalar multiplication) do not preserve the geometric structure of hyperbolic space, we require specialized mappings to transfer representations between Euclidean and hyperbolic spaces. In hyperbolic geometry, the tangent space $T_{\mathbf{x}}\mathbb{B}_c^n$ at any point $\mathbf{x} \in \mathbb{B}_c^n$ serves as a first-order approximation of $\mathbb{B}_c^n$, representing an $n$-dimensional Euclidean space that locally approximates the hyperbolic structure. The tangent space $T_{\mathbf{x}}\mathbb{B}_c^n$ and $\mathbb{B}_c^n$ are mapped to each other by exponential $(\exp_{\mathbf{x}}^{\mathbb{B},c} : T_{\mathbf{x}}\mathbb{B}_c^n \rightarrow \mathbb{B}_c^n)$ and logarithmic $(\log_{\mathbf{x}}^{\mathbb{B},c} : \mathbb{B}_c^n \rightarrow T_{\mathbf{x}}\mathbb{B}_c^n)$ maps, respectively. For $\mathbf{v} \neq \mathbf{0}$ and $\mathbf{y} \neq \mathbf{x}$, these maps are given by:

$$\exp_{\mathbf{x}}^{\mathbb{B},c}(\mathbf{v}) = \mathbf{x} \oplus_c \left( \tanh\left(\sqrt{c}\frac{\lambda_{c,\mathbf{x}}\|\mathbf{v}\|}{2}\right) \frac{\mathbf{v}}{\sqrt{c}\|\mathbf{v}\|} \right), \quad (5)$$

$$\log_{\mathbf{x}}^{\mathbb{B},c}(\mathbf{y}) = \frac{2}{\sqrt{c}\lambda_{c,\mathbf{x}}} \tanh^{-1}\left(\sqrt{c}\|\mathbf{d}\|\right) \frac{\mathbf{d}}{\|\mathbf{d}\|}, \quad (6)$$

where $\oplus_c$ denotes Möbius addition and $\mathbf{d} = (-\mathbf{x}) \oplus_c \mathbf{y}$. Detailed definitions of Möbius operations are provided in the Appendix.

## 4. Methodology

We present the proposed HypCL framework in this section.

### 4.1. Overview

Given a pre-trained CLIP model, we freeze the original visual and text encoders and introduce lightweight task-specific adapters into the visual encoder. HypCL consists

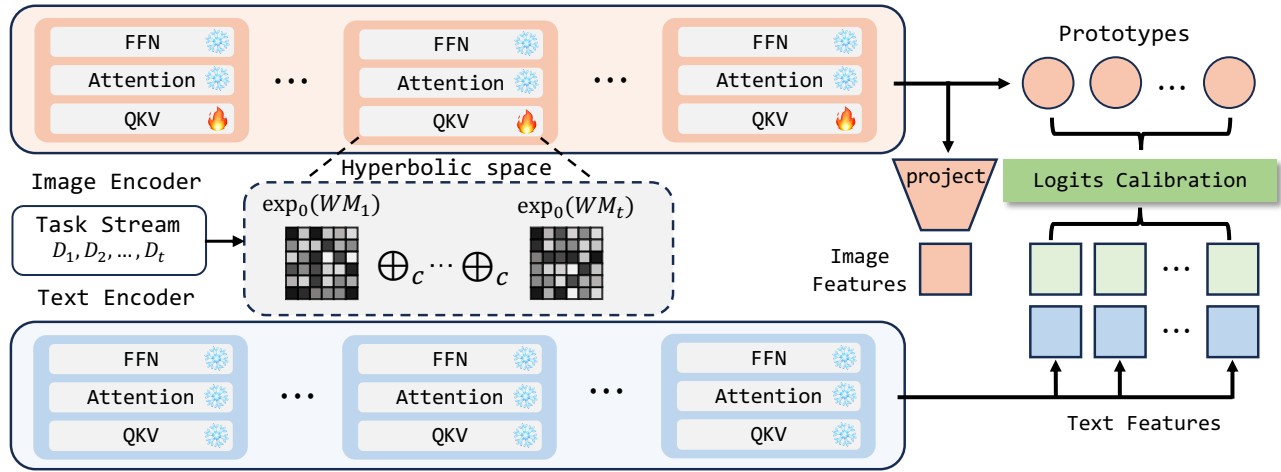

*Figure 1.* Overview of HypCL. We freeze CLIP and attach task-specific adapters to the visual encoder's attention projections. Task updates are composed in the Poincaré ball, and hyperbolic visual-text logits are fused with visual-prototype logits at inference.

of four main components. First, each task learns adapters on the query and value projections of the attention blocks, while previous task updates remain frozen. Second, instead of linearly merging these updates, HypCL treats them as tangent-space updates and composes them sequentially in the Poincaré ball. Third, HypCL uses hyperbolic distances between adapted image features and frozen text embeddings for classification. Finally, it fuses these logits with visual-prototype logits at inference. Figure 1 illustrates the overall architecture of HypCL.

### 4.2. Hyperbolic Fine-tuning

The key motivation behind our approach lies in the geometric properties of hyperbolic space. Unlike Euclidean space, hyperbolic space provides exponentially expanding capacity and is particularly well-suited for organizing representations with latent hierarchical structures (Nickel & Kiela, 2017; Hamann, 2018; Desai et al., 2023). This property naturally matches continual learning, where the model must progressively accommodate an expanding class space and distinguish categories at different semantic granularities. In CLIP-based continual learning, the frozen text encoder provides a stable semantic space, while the visual encoder must remain plastic enough to learn new classes. The challenge is to improve visual discriminability without breaking this alignment.

Our insight is to formulate visual adaptation as a geometric update around the frozen CLIP weights. A standard parameter-efficient adapter for a projection matrix $\mathbf{W}$ usually learns an update of the form $\mathbf{WM}$ and applies it through Euclidean addition:

$$\mathbf{W}_{\mathrm{euc}} = \mathbf{W} + \mathbf{WM}. \qquad (7)$$

Although this design is simple and efficient, it assumes

that task-specific corrections lie in a flat vector space and can be accumulated linearly. This assumption is restrictive in continual learning, because different tasks may require corrections along different semantic directions. Directly adding these corrections can cause interference, cancellation, or representation drift away from the pre-trained CLIP geometry.

To address this issue, HypCL replaces Euclidean addition with its hyperbolic counterpart. Inspired by HyperET (Peng et al., 2025a), we adopt the Poincaré ball model $\mathbb{B}_c^n = \{\mathbf{x} \in \mathbb{R}^n : c\|\mathbf{x}\|^2 < 1\}$ as the underlying hyperbolic space, where $c > 0$ and $-c$ denotes the curvature. The frozen projection matrix is treated as a stable anchor, while $\mathbf{WM}$ is computed from the same projection and mapped from the tangent space at the origin. Instead of merging the two terms by ordinary addition, HypCL maps them to the Poincaré ball, composes them by Möbius addition, and maps the result back to the tangent space.

For notation, we apply hyperbolic operations to matrices by viewing them as vectors in the ambient space and reshaping the results back to their original dimensions. For a frozen projection matrix $\mathbf{W}$ and a learnable adapter matrix $\mathbf{M}$, the hyperbolic adapter is defined as

$$A(\mathbf{W}; \mathbf{M}) = \log_{\mathbf{0}}^{\mathbb{B},c}\left(\exp_{\mathbf{0}}^{\mathbb{B},c}(\mathbf{W}) \oplus_c \exp_{\mathbf{0}}^{\mathbb{B},c}(\mathbf{WM})\right). \qquad (8)$$

This formulation keeps the adapted projection explicitly anchored to the frozen CLIP projection, while incorporating the adapter as a geometry-aware displacement. Therefore, the update is not an unconstrained perturbation of the pre-trained weights, but a structured transformation in a negatively curved space.

This construction also clarifies the role of curvature. When

$c \to 0$, the Poincaré ball approaches Euclidean space, the exponential and logarithmic maps at the origin reduce to identity maps, and Möbius addition reduces to ordinary vector addition. Consequently, the proposed adapter recovers the standard Euclidean adapter:

$$\lim_{c \to 0} \log_{\mathbf{0}}^{\mathbb{B},c} \left( \exp_{\mathbf{0}}^{\mathbb{B},c}(\mathbf{W}) \oplus_c \exp_{\mathbf{0}}^{\mathbb{B},c}(\mathbf{WM}) \right) = \mathbf{W} + \mathbf{WM}. \tag{9}$$

Thus, HypCL can be viewed as a geometric generalization of Euclidean adapter tuning. In the continual setting, the same principle extends naturally to a sequence of task adapters. As detailed in Section 4.3, the Euclidean limit of the multi-task formulation becomes $\mathbf{W} + \mathbf{W} \sum_{i=1}^{t} \mathbf{M}_i$, while the hyperbolic formulation composes task updates through a sequence of Möbius additions. This provides a nonlinear mechanism for modeling interactions among task updates.

Based on this formulation, we apply hyperbolic fine-tuning to the visual encoder of CLIP while keeping the text encoder frozen. The frozen text embeddings serve as stable semantic anchors throughout the continual learning process. By adapting only the visual projections through geometry-aware updates, HypCL improves downstream discriminability while reducing the risk of disrupting the cross-modal structure learned during pre-training.

For multi-head self-attention with input token sequence $\mathbf{H} \in \mathbb{R}^{L \times d}$, the query, key, and value representations are computed as $\mathbf{Q} = \mathbf{HW}_Q$, $\mathbf{K} = \mathbf{HW}_K$, and $\mathbf{V} = \mathbf{HW}_V$, where $\mathbf{W}_Q, \mathbf{W}_K, \mathbf{W}_V \in \mathbb{R}^{d \times d}$ are the projection weight matrices. We apply hyperbolic fine-tuning to the query and value projections while leaving the key projection unchanged, which improves adaptation capacity without excessively perturbing the attention structure inherited from CLIP. For a single adapter, this gives

$$\tilde{\mathbf{W}}_Q = A(\mathbf{W}_Q; \mathbf{M}_Q), \tag{10}$$
$$\tilde{\mathbf{W}}_V = A(\mathbf{W}_V; \mathbf{M}_V). \tag{11}$$

The next subsection extends this single-adapter formulation to chronological task composition.

For parameter-efficient fine-tuning, each task matrix is restricted to a banded form with bandwidth $b$:

$$(\mathbf{M}_i)_{pq} = \begin{cases} m_{pq} & \text{if } |p - q| \leq b, \\ 0 & \text{otherwise,} \end{cases} \tag{12}$$

which keeps the adapter lightweight while allowing local channel interactions for task adaptation.

### 4.3. Hyperbolic Update Composition

We now extend the single-adapter formulation to the continual setting. The following formulation applies independently to the query and value projections; for notational simplicity, we omit the projection subscript and use $\mathbf{W}$ to denote either $\mathbf{W}_Q$ or $\mathbf{W}_V$. We denote the resulting multi-task adapter after task $t$ as $A_t(\mathbf{W})$. For each projection matrix $\mathbf{W}$, every task owns one banded transformation matrix $\mathbf{M}_i$ with the same bandwidth constraint defined above. After task $i$ is trained, $\mathbf{M}_i$ is frozen and is not modified by subsequent tasks. We do not average or normalize task matrices across tasks; each learned matrix directly participates in the geometric composition.

Given the current task $t$, HypCL composes the frozen task matrices in chronological order. We first map the original projection $\mathbf{W}$ to the Poincaré ball:

$$\mathbf{P}_0 = \exp_{\mathbf{0}}^{\mathbb{B},c}(\mathbf{W}). \tag{13}$$

For task $i$, the tangent update is computed from the same original projection $\mathbf{W}$, rather than from the output of previous tasks:

$$\mathbf{P}_i = \mathbf{P}_{i-1} \oplus_c \exp_{\mathbf{0}}^{\mathbb{B},c}(\mathbf{WM}_i), \quad i = 1, \ldots, t. \tag{14}$$

After composing all task matrices up to task $t$, the adapted projection is obtained by mapping back to the tangent space:

$$A_t(\mathbf{W}) = \log_{\mathbf{0}}^{\mathbb{B},c}(\mathbf{P}_t). \tag{15}$$

This design keeps each task transformation anchored to the frozen CLIP projection while modeling interactions among tasks through sequential Möbius composition.

The corresponding Euclidean adapter would collapse all task matrices into a single linear update,

$$A_{\text{euc}}(\mathbf{W}) = \mathbf{W} + \mathbf{W} \sum_{i=1}^{t} \mathbf{M}_i. \tag{16}$$

In contrast, HypCL uses

$$A_{\text{hyp}}(\mathbf{W}) = \log_{\mathbf{0}}^{\mathbb{B},c} \left( \exp_{\mathbf{0}}^{\mathbb{B},c}(\mathbf{W}) \oplus_c \exp_{\mathbf{0}}^{\mathbb{B},c}(\mathbf{WM}_1) \right.$$
$$\left. \oplus_c \cdots \oplus_c \exp_{\mathbf{0}}^{\mathbb{B},c}(\mathbf{WM}_t) \right). \tag{17}$$

Thus the final projection is not produced by a linearly aggregated task matrix, but by a sequence of non-linear geometric updates in the Poincaré ball.

### 4.4. Hyperbolic Distance Classification

HypCL also aligns the prediction objective with the hyperbolic adapter. For each class name $y$, we construct text prompts and encode them with the frozen CLIP text encoder. The normalized prompt embeddings are averaged and normalized again to form a text embedding $\mathbf{e}_y$. Given an image $\mathbf{x}$, the adapted visual encoder produces a normalized feature $\mathbf{f}(\mathbf{x})$. We map both image and text features to a Poincaré

ball and compute logits by negative geodesic distance:

$$\mathbf{z}_x = \exp_{\mathbf{0}}^{\mathbb{B},c_d}(r\mathbf{f}(\mathbf{x})), \tag{18}$$

$$\mathbf{z}_y = \exp_{\mathbf{0}}^{\mathbb{B},c_d}(r\mathbf{e}_y), \tag{19}$$

$$\ell_y^{\mathrm{hyp}}(\mathbf{x}) = -s \cdot d_{c_d}^{\mathbb{B}}(\mathbf{z}_x, \mathbf{z}_y), \tag{20}$$

where $c_d$ is the curvature for distance logits, $r$ is a radius scale, and $s$ is a logit scale. During task $t$, the cross-entropy loss is computed only over $\mathcal{Y}_t$. At inference, the same hyperbolic distance logits are computed over $\mathcal{Y}_{\leq t}$.

To further enhance stability, we maintain visual prototypes computed in the selected visual feature space after each task. For each seen class $y$, the visual prototype $\boldsymbol{\mu}_y$ is the mean of training features belonging to that class. At inference, we apply Gaussian Discriminant Analysis (GDA) with a shared covariance to obtain visual-prototype logits $\ell_y^{\mathrm{proto}}(\mathbf{x})$, which are combined with the hyperbolic visual-text logits:

$$\boldsymbol{\ell}^{\mathrm{final}}(\mathbf{x}) = (1-\eta)Z(\boldsymbol{\ell}^{\mathrm{hyp}}(\mathbf{x})) + \eta Z(\boldsymbol{\ell}^{\mathrm{proto}}(\mathbf{x})), \tag{21}$$

where $Z(\cdot)$ denotes per-sample standardization and $\eta \in [0, 1]$ balances the two logit sources. These visual prototypes serve as stable anchors in the well-separated feature space and provide complementary calibration signals at inference.

## 5. Experiments

We evaluate HypCL on standard class-incremental learning benchmarks. This section first describes the experimental setup, then presents the main results and analyses.

### 5.1. Experimental Setup

**Datasets.** Following previous works (Zhou et al., 2025a), we evaluate on nine datasets that exhibit domain shift from CLIP's pre-training distribution. These include CIFAR100 (Krizhevsky, 2009), CUB200 (Wah et al., 2011), ObjectNet (Barbu et al., 2019), ImageNet-R (Hendrycks et al., 2021), FGVCAircraft (Maji et al., 2013), Stanford-Cars (Krause et al., 2013), Food101 (Bossard et al., 2014), SUN397 (Xiao et al., 2010), and UCF101 (Soomro et al., 2012). For consistent data splits (Zhou et al., 2025a;b), we sample 100 classes from CIFAR100, Aircraft, Cars, Food, and UCF, 200 classes from CUB200, ObjectNet, and ImageNet-R, and 300 classes from SUN. We adopt the widely used B-$m$ Inc-$n$ protocol (Zhou et al., 2024a; 2025a), where the model first learns $m$ classes and then adds $n$ new classes at each subsequent task. The class order is randomly shuffled with a fixed seed (1993) and kept consistent across all methods for fair comparison. The test set at each task contains all classes observed so far.

**Baselines and Metrics.** We compare HypCL with continual learning methods based on pre-trained models, including L2P (Wang et al., 2022b), DualPrompt (Wang et al., 2022a), CODA-Prompt (Smith et al., 2023), and SimpleCIL (Zhou et al., 2024a). We include CLIP-based methods that freeze the backbone and adapt lightweight components, such as CoOp (Zhou et al., 2022), RAPF (Huang et al., 2024), and ENGINE (Zhou et al., 2025a). We also compare our method with exemplar-based CL methods, such as iCaRL (Rebuffi et al., 2017), MEMO (Zhou et al., 2023) and PROOF (Zhou et al., 2025b). Additionally, we report zero-shot CLIP as a reference and naive fine-tuning as a lower bound. All methods use the same CLIP initialization for fair comparison. Let $\mathcal{A}_t$ denote the accuracy after task $t$ and $T$ the total number of tasks. We report the final accuracy $\mathcal{A}_T$ and the average accuracy $\bar{\mathcal{A}} = \frac{1}{T}\sum_{t=1}^{T}\mathcal{A}_t$. We additionally include per-task curves to visualize stability and plasticity.

**Implementation Details.** All experiments are conducted on NVIDIA A6000 GPUs using PyTorch. Following prior work (Zhou et al., 2025a), we use CLIP with ViT-B/16 as the backbone for all compared methods to ensure fair comparison. For vision-based methods that do not utilize textual prompts (e.g., L2P, DualPrompt, CODA-Prompt), we use CLIP's visual encoder as their initialization. We report results using LAION-400M pre-trained CLIP (Schuhmann et al., 2022). We use AdamW optimizer with a batch size of 128 to train the model. The learning rate is tuned per dataset and decays with a cosine schedule. For HypCL, we train only the current task adapters and use hyperbolic distance logits for both training and evaluation. We update the visual prototypes after each task and fuse the visual-prototype logits with hyperbolic visual-text logits during inference. For visual prototype extraction, we use two feature spaces depending on the dataset. Visual prototypes are computed with the frozen pre-trained visual backbone on ImageNet-R, CIFAR100, Food101, SUN, and UCF, and with the current adapted visual backbone on Cars, Aircraft, ObjectNet, and CUB. This choice keeps the visual prototype space stable on broad-distribution datasets while allowing visual prototypes to capture task-specific discriminative features on fine-grained datasets.

### 5.2. Experimental Results

Table 1 summarizes results on nine benchmarks. HypCL achieves the best performance in 17 out of 18 settings, consistently demonstrating superior average accuracy $\bar{\mathcal{A}}$ and final accuracy $\mathcal{A}_T$. On CIFAR100 (B0 Inc10), HypCL achieves 89.09% average accuracy and 82.95% final accuracy, surpassing Engine by +2.17% and +3.73%, respectively. The larger improvement in final accuracy indicates that HypCL effectively retains knowledge across the entire task sequence. The advantages of HypCL are particularly pronounced on fine-grained datasets. On Aircraft (B0 Inc10), HypCL achieves 71.34% average accuracy, outper-

*Table 1.* Comparison of class-incremental learning methods on nine benchmarks. We report average accuracy $\bar{A}$ and final accuracy $A_T$. Best results are highlighted in **bold**. All methods are initialized from the same pre-trained CLIP model to ensure fair comparison.

| Method | Aircraft | | | | CIFAR100 | | | | Cars | | | |
| | B0 Inc10 | | B50 Inc10 | | B0 Inc10 | | B50 Inc10 | | B0 Inc10 | | B50 Inc10 | |
| | $\bar{A}$ | $A_T$ | $\bar{A}$ | $A_T$ | $\bar{A}$ | $A_T$ | $\bar{A}$ | $A_T$ | $\bar{A}$ | $A_T$ | $\bar{A}$ | $A_T$ |
|---|---|---|---|---|---|---|---|---|---|---|---|---|
| Finetune | 3.16 | 0.96 | 1.72 | 1.05 | 7.84 | 4.44 | 5.30 | 2.46 | 3.14 | 1.10 | 1.54 | 1.13 |
| CoOp | 14.54 | 7.14 | 13.05 | 7.77 | 47.00 | 24.24 | 41.23 | 24.12 | 36.46 | 21.65 | 37.40 | 20.87 |
| SimpleCIL | 59.24 | 48.09 | 53.05 | 48.09 | 84.15 | 76.63 | 80.20 | 76.63 | 92.04 | 86.85 | 88.96 | 86.85 |
| ZS-CLIP | 26.66 | 17.22 | 21.70 | 17.22 | 81.81 | 71.38 | 76.49 | 71.38 | 82.60 | 76.37 | 78.32 | 76.37 |
| L2P | 47.19 | 28.29 | 44.07 | 32.13 | 82.74 | 73.03 | 81.14 | 73.61 | 76.63 | 61.82 | 76.37 | 65.64 |
| DualPrompt | 44.30 | 25.83 | 46.07 | 33.57 | 81.63 | 72.44 | 80.12 | 72.57 | 76.26 | 62.94 | 76.88 | 67.55 |
| CODA-Prompt | 45.98 | 27.69 | 45.14 | 32.28 | 82.43 | 73.43 | 78.69 | 71.58 | 80.21 | 66.47 | 75.06 | 64.19 |
| RAPF | 50.38 | 23.61 | 40.47 | 25.44 | 86.14 | 78.04 | 82.17 | 77.93 | 82.89 | 62.85 | 75.87 | 63.19 |
| Engine | 69.69 | 58.69 | 64.38 | 59.02 | 86.92 | 79.22 | 83.15 | 79.47 | 94.14 | 90.08 | 91.61 | 90.03 |
| Ours | **71.34** | **60.64** | **67.24** | **62.38** | **89.09** | **82.95** | **86.87** | **83.42** | **94.44** | **90.79** | **92.33** | **91.08** |

| Method | ImageNet-R | | | | CUB | | | | UCF | | | |
| | B0 Inc20 | | B100 Inc20 | | B0 Inc20 | | B100 Inc20 | | B0 Inc10 | | B50 Inc10 | |
| | $\bar{A}$ | $A_T$ | $\bar{A}$ | $A_T$ | $\bar{A}$ | $A_T$ | $\bar{A}$ | $A_T$ | $\bar{A}$ | $A_T$ | $\bar{A}$ | $A_T$ |
|---|---|---|---|---|---|---|---|---|---|---|---|---|
| Finetune | 1.37 | 0.43 | 1.01 | 0.88 | 2.06 | 0.64 | 0.56 | 0.47 | 4.51 | 1.59 | 1.21 | 0.80 |
| CoOp | 60.73 | 37.52 | 54.20 | 39.77 | 27.61 | 8.57 | 24.03 | 10.14 | 47.85 | 33.46 | 42.02 | 24.74 |
| SimpleCIL | 81.06 | 74.48 | 76.84 | 74.48 | 83.81 | 77.52 | 79.75 | 77.52 | 90.44 | 85.68 | 88.12 | 85.68 |
| ZS-CLIP | 83.37 | 77.17 | 79.57 | 77.17 | 74.38 | 63.06 | 67.96 | 63.06 | 75.50 | 67.64 | 71.44 | 67.64 |
| L2P | 75.97 | 66.52 | 72.82 | 66.77 | 70.87 | 57.93 | 75.64 | 66.12 | 86.34 | 76.43 | 83.95 | 76.62 |
| DualPrompt | 76.21 | 66.65 | 73.22 | 67.58 | 69.89 | 57.46 | 74.40 | 64.84 | 85.21 | 75.82 | 84.31 | 76.35 |
| CODA-Prompt | 77.69 | 68.95 | 73.71 | 68.05 | 73.12 | 62.98 | 73.95 | 62.21 | 87.76 | 80.14 | 83.04 | 75.03 |
| RAPF | 81.26 | 70.48 | 76.10 | 70.23 | 79.09 | 62.77 | 72.82 | 62.93 | 92.28 | 80.33 | 90.31 | 81.55 |
| Engine | 86.22 | 80.37 | 83.63 | 80.98 | 86.65 | 80.20 | 82.59 | 79.30 | **94.35** | **90.03** | 92.51 | 89.58 |
| Ours | **88.98** | **83.67** | **86.31** | **84.18** | **86.83** | **80.83** | **83.32** | **80.79** | 93.99 | 89.88 | **93.55** | **90.68** |

| Method | SUN | | | | Food | | | | ObjectNet | | | |
| | B0 Inc30 | | B150 Inc30 | | B0 Inc10 | | B50 Inc10 | | B0 Inc20 | | B100 Inc20 | |
| | $\bar{A}$ | $A_T$ | $\bar{A}$ | $A_T$ | $\bar{A}$ | $A_T$ | $\bar{A}$ | $A_T$ | $\bar{A}$ | $A_T$ | $\bar{A}$ | $A_T$ |
|---|---|---|---|---|---|---|---|---|---|---|---|---|
| Finetune | 4.51 | 1.59 | 0.78 | 0.72 | 3.49 | 1.71 | 2.14 | 1.52 | 1.34 | 0.47 | 0.69 | 0.54 |
| CoOp | 45.93 | 23.11 | 39.33 | 24.89 | 36.01 | 14.18 | 33.13 | 18.67 | 21.24 | 6.29 | 16.21 | 6.82 |
| SimpleCIL | 82.13 | 75.58 | 78.62 | 75.58 | 87.89 | 81.65 | 84.73 | 81.65 | 52.06 | 40.13 | 45.11 | 40.13 |
| ZS-CLIP | 79.42 | 72.11 | 74.95 | 72.11 | 87.86 | 81.92 | 84.75 | 81.92 | 38.43 | 26.43 | 31.12 | 26.43 |
| L2P | 82.82 | 74.54 | 79.57 | 73.10 | 85.66 | 77.33 | 80.42 | 73.13 | 51.40 | 39.39 | 48.91 | 42.83 |
| DualPrompt | 82.46 | 74.40 | 79.37 | 73.02 | 84.92 | 77.29 | 80.00 | 72.75 | 52.62 | 40.72 | 49.08 | 42.92 |
| CODA-Prompt | 83.34 | 75.71 | 80.38 | 74.17 | 86.18 | 78.78 | 80.98 | 74.13 | 46.49 | 34.13 | 40.57 | 34.13 |
| RAPF | 82.13 | 72.47 | 78.04 | 73.10 | 88.57 | 81.15 | 85.53 | 81.17 | 48.67 | 27.43 | 39.28 | 28.73 |
| Engine | 85.04 | 78.54 | 81.57 | 78.45 | 89.81 | 83.89 | 86.89 | 83.94 | 59.11 | 45.19 | 51.32 | 44.99 |
| Ours | **85.85** | **80.00** | **83.57** | **80.31** | **90.07** | **84.67** | **87.61** | **84.80** | **62.31** | **52.07** | **55.95** | **51.78** |

forming Engine by +1.65%. On CUB (B0 Inc20), HypCL attains 86.83% average accuracy and 80.83% final accuracy. These results support our hypothesis that hyperbolic geometry better accommodates fine-grained class distinctions by providing sufficient space for closely related categories. On domain-shifted benchmarks, HypCL also demonstrates strong adaptability: on ImageNet-R (B0 Inc20), HypCL achieves 88.98% average accuracy and 83.67% final accuracy (+2.76% and +3.30% over Engine); on ObjectNet, HypCL shows substantial gains with 62.31% average accuracy (+3.20% over Engine).

Figure 2 presents the per-task accuracy curves. As new classes are introduced, all methods exhibit accuracy degradation due to the expanded label space. However, HypCL consistently maintains higher accuracy with a slower decline rate. The gradual accuracy degradation, rather than abrupt drops, indicates that the hyperbolic update composition effectively balances new knowledge integration with the preservation of prior representations.

We compare HypCL with exemplar-based CIL methods, including iCaRL (Rebuffi et al., 2017), MEMO (Zhou et al., 2023), and PROOF (Zhou et al., 2025b), which store 20 exemplars per class to replay historical samples during training. As shown in Table 2, despite not using any exemplars,

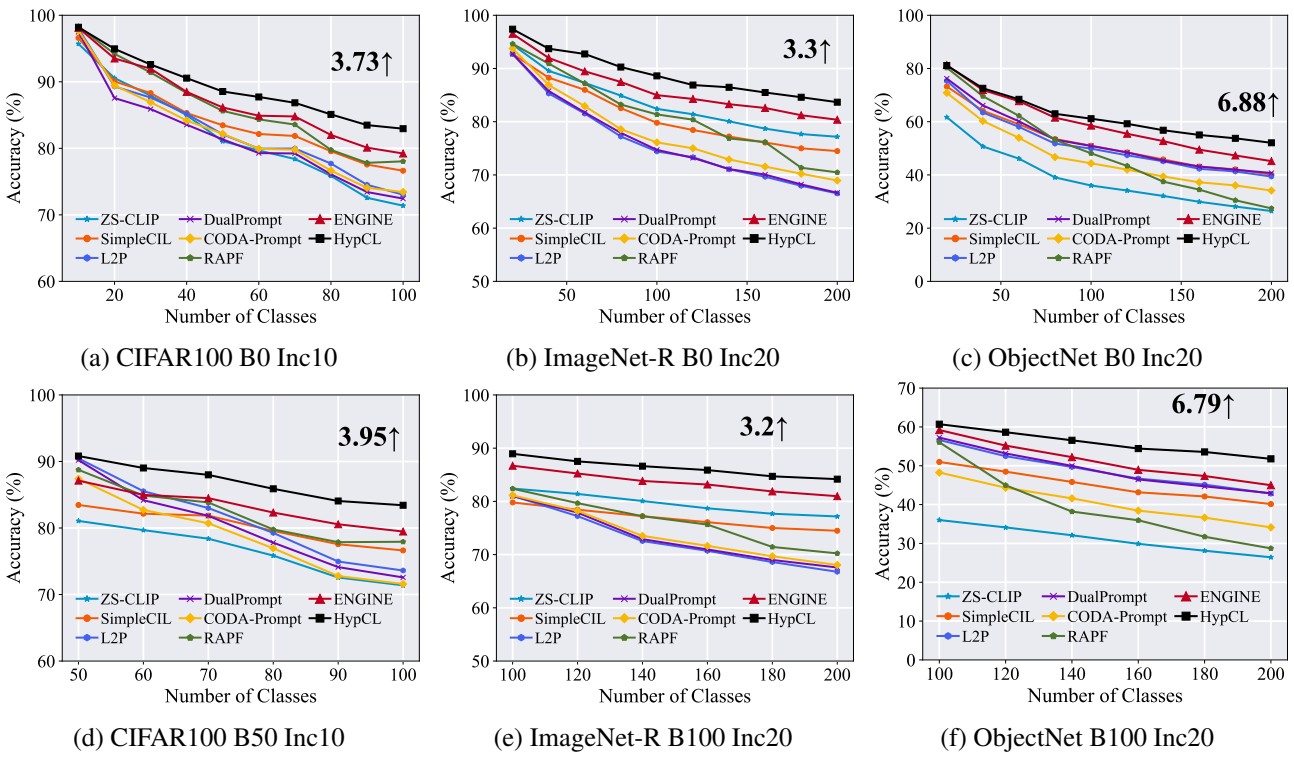

*Figure 2.* Per-task accuracy curves across incremental tasks on three benchmarks under both B0 and B-$m$ protocols. HypCL consistently maintains higher accuracy throughout the learning process, demonstrating its effectiveness in mitigating forgetting while adapting to new classes.

*Table 2.* Comparison with traditional exemplar-based CIL methods. HypCL does not use any exemplars.

| Method | Exemplars | Aircraft B0 Inc10 | | SUN B0 Inc30 | |
|---|---|---|---|---|---|
| | | $\bar{\mathcal{A}}$ | $\mathcal{A}_T$ | $\bar{\mathcal{A}}$ | $\mathcal{A}_T$ |
| iCaRL | 20 / class | 53.60 | 43.98 | 78.56 | 67.30 |
| MEMO | 20 / class | 42.24 | 25.41 | 81.48 | 73.45 |
| PROOF | 20 / class | 61.00 | 53.59 | 83.57 | 77.28 |
| HypCL | 0 | **71.34** | **60.64** | **85.85** | **80.00** |

*Table 3.* Ablation study on ImageNet-R and CIFAR100. We report final accuracy $\mathcal{A}_T$ and average accuracy $\bar{\mathcal{A}}$.

| Dataset | Variant | $\mathcal{A}_T$ | $\bar{\mathcal{A}}$ |
|---|---|---|---|
| ImageNet-R | Euclidean | 81.02 | 85.88 |
| | HypCL w/o prototype | 83.00 | 87.84 |
| | HypCL + prototype | 83.67 | 88.98 |
| CIFAR100 | Euclidean | 74.25 | 85.09 |
| | HypCL w/o prototype | 79.77 | 87.71 |
| | HypCL + prototype | 82.95 | 89.09 |

HypCL significantly outperforms all exemplar-based methods on both Aircraft and SUN benchmarks. On Aircraft (B0 Inc10), HypCL achieves 71.34% average accuracy and 60.64% final accuracy, surpassing the best exemplar-based method by +10.34% and +7.05%, respectively. This substantial margin demonstrates that leveraging the geometric properties of hyperbolic space combined with CLIP's pretrained representations provides a more effective approach to mitigating forgetting than naive exemplar replay. The results highlight the advantage of HypCL, which achieves superior performance while avoiding the storage and privacy costs associated with exemplar-based methods.

## 5.3. Ablation Study

We conduct ablation studies on ImageNet-R (B0 Inc20) and CIFAR100 (B0 Inc10). Table 3 compares a Euclidean

adapter baseline with cosine logits, HypCL without visual prototype calibration, and the variant with visual prototype calibration. The first comparison evaluates the effect of hyperbolic update composition and hyperbolic distance logits, while the second comparison evaluates visual prototype calibration on top of them.

Without visual prototype calibration, replacing Euclidean updates and cosine logits with hyperbolic update composition and hyperbolic distance logits improves final accuracy by 1.98% on ImageNet-R and 5.52% on CIFAR100. Adding visual prototype calibration further improves both datasets, with a larger gain on CIFAR100. This indicates that visual prototype calibration is most useful when the class space is more sensitive to incremental drift.

*Table 4.* Sensitivity analysis of bandwidth $b$ and visual proto-type fusion weight $\eta$ on ImageNet-R (B0 Inc20). We report final accuracy $\mathcal{A}_T$ and average accuracy $\bar{\mathcal{A}}$.

| $b$ | 1 | 3 | 5 | 7 | 9 |
|---|---|---|---|---|---|
| $\mathcal{A}_T$ | 83.63 | 83.37 | 83.67 | 83.43 | 83.33 |
| $\bar{\mathcal{A}}$ | 88.39 | 88.70 | 88.98 | 88.99 | 88.83 |
| $\eta$ | 0.4 | 0.5 | 0.6 | 0.7 | 0.8 |
| $\mathcal{A}_T$ | 83.65 | 83.67 | 83.60 | 83.33 | 82.33 |
| $\bar{\mathcal{A}}$ | 88.89 | 88.98 | 88.82 | 88.41 | 87.63 |

### 5.4. Sensitivity Analysis

We analyze the sensitivity of HypCL to two key hyperparameters: the bandwidth $b$ of the banded adapter and the visual prototype fusion weight $\eta$. The bandwidth controls adapter expressiveness, while $\eta$ balances hyperbolic visual-text logits and visual-prototype logits at inference. Table 4 reports results on ImageNet-R (B0 Inc20). When varying $b$, we fix $\eta = 0.5$; when varying $\eta$, we fix $b = 5$.

For bandwidth $b$, performance is stable across a moderate range of values. Increasing $b$ from 1 to 5 improves average accuracy from 88.39% to 88.98% and gives the best final accuracy of 83.67%. Further increasing the bandwidth does not bring clear gains: $b = 7$ gives a marginally higher average accuracy but lower final accuracy. This suggests that a moderate bandwidth is sufficient for adaptation.

For the fusion weight $\eta$, the best final and average accuracy are both obtained at $\eta = 0.5$, reaching 83.67% and 88.98%, respectively. A smaller value underuses visual prototype calibration, while larger values overemphasize the visual-prototype head: when $\eta$ increases to 0.8, final accuracy drops to 82.33% and average accuracy drops to 87.63%. This confirms that visual-prototype logits are most effective as a complementary signal to the hyperbolic visual-text logits.

## 6. Conclusion

We propose HypCL, a parameter-efficient framework that adapts CLIP in hyperbolic space for continual learning. By leveraging the exponential volume growth of hyperbolic geometry, HypCL accommodates expanding class spaces without compressing prior representations. The hierarchical inductive bias of hyperbolic space further complements the semantic structure in CLIP's cross-modal embeddings. The framework learns task-specific adapters, composes their updates with Möbius addition, and aligns prediction with the geometry through hyperbolic visual-text distances. Visual-prototype logit fusion further improves inference over all seen classes. Extensive experiments demonstrate that HypCL consistently outperforms existing CLIP-based continual learning methods across diverse benchmarks.

## Acknowledgements

This work was partially supported by NSFC (U23A20382), the Fundamental and Interdisciplinary Disciplines Breakthrough Plan of the Ministry of Education of China (No. JYB2025XDXM118), and the Fundamental Research Funds for the Central Universities (2026300271).

## Impact Statement

This paper presents work whose goal is to advance the field of Machine Learning. There are many potential societal consequences of our work, none which we feel must be specifically highlighted here.

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

# A. Hyperbolic Operations

This appendix provides formal definitions of the hyperbolic operations used throughout this work, including Möbius addition, Möbius multiplication, and the exponential and logarithmic maps.

**Möbius addition.** For two points $\mathbf{x}, \mathbf{y} \in \mathbb{B}_c^n$, the Möbius addition is defined as:

$$\mathbf{x} \oplus_c \mathbf{y} := \frac{(1 + 2c\langle \mathbf{x}, \mathbf{y} \rangle + c\|\mathbf{y}\|^2)\mathbf{x} + (1 - c\|\mathbf{x}\|^2)\mathbf{y}}{1 + 2c\langle \mathbf{x}, \mathbf{y} \rangle + c^2\|\mathbf{x}\|^2\|\mathbf{y}\|^2}. \tag{22}$$

**Möbius scalar multiplication.** For a scalar $r \in \mathbb{R}$ and a point $\mathbf{x} \in \mathbb{B}_c^n$, the Möbius scalar multiplication is defined as:

$$r \otimes_c \mathbf{x} := \frac{1}{\sqrt{c}} \tanh\left(r \cdot \tanh^{-1}(\sqrt{c}\|\mathbf{x}\|)\right) \frac{\mathbf{x}}{\|\mathbf{x}\|}. \tag{23}$$

**Möbius matrix multiplication.** For a matrix $\mathbf{M} \in \mathbb{R}^{n \times n}$ and a point $\mathbf{x} \in \mathbb{B}_c^n$ with $\mathbf{Mx} \neq \mathbf{0}$, the Möbius matrix multiplication is defined as:

$$\mathbf{M} \otimes_c \mathbf{x} := \frac{1}{\sqrt{c}} \tanh\left(\frac{\|\mathbf{Mx}\|}{\|\mathbf{x}\|} \cdot \tanh^{-1}(\sqrt{c}\|\mathbf{x}\|)\right) \frac{\mathbf{Mx}}{\|\mathbf{Mx}\|}. \tag{24}$$

**Tangent space and mappings.** The tangent space $T_{\mathbf{x}}\mathbb{B}_c^n$ at a point $\mathbf{x} \in \mathbb{B}_c^n$ provides a local Euclidean approximation of the hyperbolic manifold $\mathbb{B}_c^n$. The tangent space and the Poincaré ball are connected via the exponential map ($\exp_{\mathbf{x}}^{\mathbb{B},c} : T_{\mathbf{x}}\mathbb{B}_c^n \to \mathbb{B}_c^n$) and the logarithmic map ($\log_{\mathbf{x}}^{\mathbb{B},c} : \mathbb{B}_c^n \to T_{\mathbf{x}}\mathbb{B}_c^n$). For $\mathbf{v} \in T_{\mathbf{x}}\mathbb{B}_c^n$ with $\mathbf{v} \neq \mathbf{0}$ and $\mathbf{y} \in \mathbb{B}_c^n$ with $\mathbf{y} \neq \mathbf{x}$, these maps are given by:

$$\exp_{\mathbf{x}}^{\mathbb{B},c}(\mathbf{v}) = \mathbf{x} \oplus_c \left(\tanh\left(\frac{\sqrt{c}\lambda_{c,\mathbf{x}}\|\mathbf{v}\|}{2}\right) \frac{\mathbf{v}}{\sqrt{c}\|\mathbf{v}\|}\right), \tag{25}$$

$$\log_{\mathbf{x}}^{\mathbb{B},c}(\mathbf{y}) = \frac{2}{\sqrt{c}\lambda_{c,\mathbf{x}}} \tanh^{-1}\left(\sqrt{c}\| - \mathbf{x} \oplus_c \mathbf{y}\|\right) \frac{-\mathbf{x} \oplus_c \mathbf{y}}{\| - \mathbf{x} \oplus_c \mathbf{y}\|}, \tag{26}$$

where $\lambda_{c,\mathbf{x}} = \frac{2}{1-c\|\mathbf{x}\|^2}$ is the conformal factor.

