# OpenReview forum: "HypCL: Adapting CLIP in Hyperbolic Space for Continual Learning"
_ICML.cc/2026/Conference — ICML 2026 regular_

### Official Review · Reviewer_mTEF · 2026-02-19

**Soundness:** 3
**Presentation:** 3
**Significance:** 2
**Originality:** 2
**Overall Recommendation:** 4
**Confidence:** 4

**Summary:**

This paper proposes HypCL, a parameter-efficient continual learning framework for CLIP-based class-incremental learning. The method freezes CLIP’s original parameters and introduces task-specific hyperbolic transformation matrices that modify the query/value projection weights in the visual encoder via exp/log maps and Möbius matrix multiplication, then maps back to Euclidean weights for standard attention computation. For continual learning across tasks, HypCL maintains a task-specific $M_t$ and aggregates them using a magnitude-direction decomposition, where historical directions are frozen and historical magnitudes remain learnable. Finally, the method maintains class prototypes computed from adapted features after each task and uses them as anchors for inference-time calibration. Experiments have demonstrated the effectiveness of the proposed method.

**Compliance With Llm Reviewing Policy:**

Affirmed.

**Final Justification:**

I thank the authors for their rebuttal and clarifications. The response addresses my concerns to a certain extent. I will adjust my score to weak accept.

**Key Questions For Authors:**

1. Your continual aggregation (Eq.15–16) is structurally very similar to SD-LoRA’s magnitude–direction decoupling and additive aggregation across tasks. Could you clearly delineate what is new beyond applying a known aggregation template to a different PEFT module?
2. In my view, leveraging an existing technique is not inherently a weakness, but you might discuss why this particular technology is coupled with your approch. Did you try more geometry-consistent alternatives (e.g., composing transformations, or averaging in a tangent space via log/exp)? If not, can you provide justification or evidence that your current aggregation behaves like a valid approximation?
3. How do you ensure older prototypes remain compatible with the current feature extractor? Please provide an analysis: e.g., similarity/distance between old-class features and their stored prototypes over time. Also, it is also helpful to report trajectories of $𝛼_t$ during later tasks to verify that dynamic re-weighting is actually active.
4. Given the paper’s continual aggregation and stability claims, I would like to see stronger evidence on longer task sequences.

If the author can address these weaknesses and questions, I would be happy to raise my score.

**Limitations:**

Authors may appropriately acknowledge the limitations of their methods, such as addressing the weaknesses and questions raised above.

**Strengths And Weaknesses:**

Strengths
1. The paper presents a key insight: they attribute the increasing crowding difficulty during incremental learning to the slow growth of Euclidean space capacity. They propose using the exponential expansion of hyperbolic space to accommodate the ever-expanding category space.
2. Clear architecture choice. Adapting Q/V projections in attention while keeping the text encoder frozen is a reasonable, lightweight way to modify representation.
3. Efficiency design. The banded parameterization reduces parameter count from $O(d^2)$ to $O(bd)$, which is sensible for PEFT-style CL.
4. Reported gains across many benchmarks suggest the approach is competitive.

Weaknesses
1. Originality / positioning of the aggregation mechanism is unclear.
The proposed magnitude-direction decomposition and additive aggregation appear highly aligned with prior LoRA aggregation work, notably SD-LoRA[1]. SD-LoRA explicitly decomposes LoRA weights into magnitude and normalized direction, freezes past directions, and learns magnitudes with an additive mixture across tasks.  HypCL uses essentially the same pattern on $M_t$ (Eq.15–16). As a result, the novelty may concentrate mainly in “injecting hyperbolic geometry into the PEFT module,” and the paper should explicitly acknowledge/compare/disentangle these contributions.
2. Geometry motivation vs implementation: the “hyperbolic capacity” story may not match what is actually optimized/used.
The method introduces hyperbolic mappings for weight transformation, but the network ultimately computes standard Euclidean features and Euclidean similarity-based logits (and the hyperbolic operations serve as a parameterization/regularization of weights). This makes it less clear whether the claimed “exponential capacity” advantage is truly responsible for the gains, or whether similar gains could be obtained by a Euclidean re-parameterization of comparable expressivity.
3. Aggregation step seems geometrically inconsistent with the stated hyperbolic rationale.
The paper states that Euclidean operations (addition/scalar multiplication) do not preserve hyperbolic structure and motivates exp/log mappings accordingly. However, the continual aggregation of transformations is done via Euclidean addition $∑𝛼_tM_t$. This is plausible as a heuristic in parameter space. However, considering that this continual aggregation strategy itself has already been studied, the author might focus on discussing why this strategy is equally applicable to hyperbolic mapping matrices $M_t$.
4. Prototype calibration under a changing aggregated transformation may be internally inconsistent.
Prototypes are computed after each task using adapted features , but as training proceeds, the aggregated transformation changes because historical magnitudes remain learnable. This suggests feature space drift, where earlier prototypes may no longer correspond to the current feature extractor. The paper does not clearly describe how this mismatch is handled, nor does it provide an analysis showing it is negligible. A related possibility is that $𝛼_t$ barely changes in practice, which would weaken the claimed benefit of dynamic re-weighting.

[1] SD-loRA: Scalable decoupled low-rank adaptation for class incremental learning.

---

> ### Author Rebuttal · Authors · 2026-03-31
>
> Thank you for the detailed feedback. We have carefully considered your concerns and provide details below.
>
> **W1&Q1. What is genuinely new relative to SD-LoRA?**
>
> **R1.** The novelty of HypCL lies in where magnitude-direction decomposition is introduced and how it interacts with the rest of the framework. HypCL applies continual aggregation to hyperbolic Q/V adaptation inside CLIP visual attention, rather than to a standard Euclidean low-rank PEFT module. The adapted module controls the radial organization of representations through hyperbolic transformations, while the text encoder remains frozen as a semantic anchor. HypCL is also coupled with prototype-based calibration in exemplar-free CIL. The contribution is therefore the integration of hyperbolic weight adaptation, continual aggregation, and CLIP-anchored prototype calibration into one continual-learning pipeline. We will revise the paper to position this more precisely and avoid overstating novelty at the level of the decomposition alone.
>
> This integration is also natural in our design: the hyperbolic module improves feature organization for expanding class spaces, the aggregation mechanism preserves historical adaptation patterns, and the calibration branch exploits the resulting structure at inference time. We therefore view the method as a new combination tailored to CLIP-based exemplar-free CIL.
>
> ---
>
> **W2&Q2. The geometry-based motivation may not align fully with the implementation, since the network still computes Euclidean features/logits and the aggregation in Eq. (15)–(16) is not a geometrically exact manifold composition rule. Did the authors try more geometry-consistent alternatives, and if not, why is the current aggregation a valid approximation?**
>
> **R2.** Hyperbolic geometry in HypCL is not intended to make the full inference pipeline end-to-end hyperbolic. Instead, hyperbolic operations parameterize how the visual attention weights are adapted. After this adaptation, the network still produces CLIP-compatible Euclidean features and standard CLIP-style logits. This is deliberate, since it preserves compatibility with frozen text embeddings and the original CLIP decision space.
>
> Likewise, Eq. (15)–(16) should be understood as an aggregation rule in parameter space, not as a claim of exact manifold composition. Each task-specific transformation is decomposed into a direction and a magnitude; historical directions are frozen, while only magnitudes remain learnable. This gives a simple and stable incremental update mechanism. We therefore do not claim that Eq. (15)–(16) is the unique or exact hyperbolic composition rule, but rather that it is an effective approximation for PEFT-style continual adaptation, as supported by the long-sequence results and ablations.
>
> ---
>
> **W3&Q3. How is feature–prototype compatibility maintained over time?**
>
> **R3.** The feature extractor does change over time, but the change is constrained rather than arbitrary. Historical directions are frozen and only magnitudes are updated, so the representation evolves mainly through smooth reweighting of previously learned task-specific directions instead of repeatedly overwriting the full transformation. This helps maintain compatibility between old prototypes and the current feature space.
>
> Empirically, the calibration branch remains beneficial after continual updates. On ImageNet-R, removing the calibration (`stat=0`) reduces AvgAcc from **88.437 to 86.536** for B0Inc20 and from **85.832 to 84.063** for B100Inc20. This indicates that the stored prototypes remain compatible enough with the current feature extractor to improve prediction after multiple continual updates.
>
> ---
>
> **W4&Q4. Can you provide stronger evidence on longer task sequences?**
>
> **R4.** We added longer-sequence experiments on three datasets. We also monitored the aggregation magnitudes. As the number of alpha coefficients increases (e.g., **144 → 264 → 504** on ImageNet-R), their L2 norm remains around 137.3, and the gradient L2 stays small, so we do not observe clear optimization instability. This is consistent with the aggregation design: each new task mainly adjusts a limited number of magnitudes on top of frozen historical directions, rather than re-learning all previous transformations jointly.
>
> **Long-sequence summary**
>
> | Dataset | Protocol | AvgAcc |
> |---|---|---:|
> | CIFAR | B0Inc20 | 88.164 |
> | CIFAR | B0Inc10 | 88.999 |
> | CIFAR | B0Inc5 | 88.969 |
> | CIFAR | B50Inc10 | 85.803 |
> | CIFAR | B50Inc5 | 85.634 |
> | ImageNet-R | B100Inc20 | 85.828 |
> | ImageNet-R | B100Inc10 | 85.555 |
> | ImageNet-R | B100Inc5 | 85.321 |
> | CUB | B100Inc20 | 82.520 |
> | CUB | B100Inc10 | 82.473 |
> | CUB | B100Inc5 | 82.225 |
>
> These results suggest that the method remains stable as the number of tasks grows.
>
> HypCL integrates hyperbolic adapter parameterization, continual aggregation, and prototype calibration in a CLIP-based exemplar-free CIL framework. We thank the reviewer for helping clarify the method’s positioning.

---

> > ### Author Rebuttal · Reviewer_mTEF · 2026-04-02
> >
> > I thank the authors for their rebuttal and clarifications. The response addresses my concerns to a certain extent. I will adjust my score to weak accept.

---

> > > ### Author Response · Authors · 2026-04-03
> > >
> > > We appreciate your thoughtful review and the time you took to consider our rebuttal and clarifications. We are encouraged by your positive adjustment and will carefully incorporate your feedback to further strengthen the paper.

---

### Official Review · Reviewer_qiyU · 2026-03-09

**Soundness:** 2
**Presentation:** 3
**Significance:** 2
**Originality:** 2
**Overall Recommendation:** 3
**Confidence:** 3

**Summary:**

This paper studies continual learning with vision-language models, focusing on adapting CLIP under the class-incremental learning setting. The authors propose HypCL, a parameter-efficient framework that performs adaptation in hyperbolic space by applying Möbius transformations to the visual encoder’s attention weights while keeping the text encoder frozen. The method maintains task-specific transformations and aggregates them using a magnitude–direction decomposition to support continual learning. Additionally, class prototypes are introduced to calibrate predictions during inference. Experiments on several class-incremental benchmarks show consistent improvements over existing CLIP-based continual learning methods.

**Compliance With Llm Reviewing Policy:**

Affirmed.

**Key Questions For Authors:**

- One question concerns the role of hyperbolic geometry in the observed performance gains. Could the authors provide additional ablation studies comparing the proposed hyperbolic transformation with a similar adaptation mechanism implemented in Euclidean space?

- Also have a question about scalability when the number of tasks becomes large. Since the method maintains task-specific transformation matrices, how does the approach behave when the number of tasks increases significantly?

- The paper combines CLIP logits with prototype-based calibration during inference. Could the authors provide more analysis on how much of the improvement comes from the prototype component versus the hyperbolic adaptation itself?

**Limitations:**

Yes

**Strengths And Weaknesses:**

**Strenghts**

One strength of the paper is that it explores the use of hyperbolic geometry for continual learning with vision-language models, which is a relatively less explored direction in this area. The proposed framework is also parameter-efficient since the CLIP backbone remains frozen and only lightweight hyperbolic transformations are learned. Another positive aspect is that the method design is relatively simple and easy to integrate into existing CLIP-based pipelines.

**Weaknesses**

- A main concern is that the conceptual novelty of the approach appears somewhat limited. The method essentially combines three existing ideas: *CLIP-based continual learning*, *hyperbolic representation learning*, and *prototype-based calibration*. While the combination is reasonable, it is not entirely clear whether the overall framework introduces fundamentally new insights beyond adapting existing techniques.

- Another concern is the motivation for using hyperbolic space. Although the paper argues that hyperbolic geometry provides better capacity for representing an expanding class space, it is unclear whether this property is truly critical in practice, especially given the already high dimensionality of CLIP embeddings. The paper would benefit from stronger empirical evidence demonstrating that the improvements specifically come from the hyperbolic formulation rather than other components of the method.

- The technical contributions are somewhat incremental. The proposed transformations and aggregation mechanism are relatively straightforward extensions of standard parameter-efficient fine-tuning ideas, and the paper lacks deeper theoretical analysis or insights into why the proposed approach works particularly well for continual learning.

---

> ### Author Rebuttal · Authors · 2026-03-31
>
> We sincerely appreciate your valuable feedback. We have carefully considered your concerns and provide details below.
>
> **W1&Q1. The motivation for using hyperbolic space is not yet fully convincing. Could the authors provide stronger evidence that the hyperbolic formulation itself matters?**
>
> **R1.** We agree that the role of the hyperbolic component should be stated more precisely. We clarify this point from the paper’s sensitivity analysis. In HypCL, the hyperbolic component is realized through hyperbolic Q/V adaptation with a banded transformation matrix. When the bandwidth is highly restricted, the average accuracy on ImageNet-R is **87.29** and the final accuracy is **82.15**. As the bandwidth increases, performance improves steadily, reaching **88.72** average accuracy and **84.05** final accuracy at `b=4`, i.e., gains of about **+1.43** in average accuracy and **+1.90** in final accuracy. This indicates that the benefit does not come merely from inserting extra parameters, but from allowing a more expressive hyperbolic transformation.
>
> This is also consistent with the role of hyperbolic adaptation in our method. Hyperbolic geometry provides exponentially expanding capacity and a hierarchical inductive bias, which suit class-incremental learning with an expanding class space. In our implementation, hyperbolic Q/V adaptation adjusts the radial distribution of representations while keeping the text encoder fixed as a stable semantic anchor. Therefore, increasing the expressiveness of the hyperbolic transformation improves the model’s ability to organize newly arriving classes without severely interfering with previous ones.
>
> We also note that the calibration weight affects performance. Around the paper setting, the best results occur near **λ=0.5–0.6**, while overly small or large calibration weights are weaker. This supports our view that HypCL works because hyperbolic adaptation and calibration are complementary, rather than because of either component alone.
>
> ---
>
> **W2&Q3. The paper should analyze how much improvement comes from the prototype-based calibration component versus the hyperbolic adaptation itself.**
>
> **R2.** We agree that disentangling these components is important. The calibration term is clearly useful: in our additional ablations on ImageNet-R, removing calibration reduces performance by about **1.8–1.9 points** under the paper setting. At the same time, the hyperbolic adaptation also matters: restricting the hyperbolic transformation to the least expressive case (`b=0`) already lowers performance relative to stronger settings (`b=3–4`). Thus, the gain should not be attributed to calibration alone.
>
> The two components play different roles. Hyperbolic adaptation shapes the feature geometry for an expanding class space, while prototype-based calibration refines logits at inference time. Thus, the hyperbolic module improves the adapted representation space, and the prototype term exploits that structure. This is why they are complementary rather than redundant.
>
> ---
>
> **W3&Q2. How does the method scale when the number of tasks becomes large, given the need to maintain task-specific transformation matrices?**
>
> **R3.** We added longer-sequence experiments and observed only mild degradation:
>
> | Dataset | Protocol | AvgAcc |
> |---|---:|---:|
> | CIFAR | B0Inc20 | 88.164 |
> | CIFAR | B0Inc10 | 88.999 |
> | CIFAR | B0Inc5 | 88.968 |
> | ImageNet-R | B100Inc20 | 85.828 |
> | ImageNet-R | B100Inc10 | 85.555 |
> | ImageNet-R | B100Inc5 | 85.321 |
> | CUB | B100Inc20 | 82.520 |
> | CUB | B100Inc10 | 82.473 |
> | CUB | B100Inc5 | 82.225 |
>
> This is consistent with our aggregation design: each task contributes its own transformation, while historical knowledge is preserved through magnitude-direction decomposition. Historical directions are frozen and only magnitudes remain learnable, so new tasks mainly reweight previous directions rather than overwrite them. This reduces interference and explains the stable long-sequence behavior.
>
> ---
>
> **W4. The conceptual novelty appears limited, and the original claim about hyperbolic geometry may be too strong given the role of calibration.**
>
> **R4.** We agree that the claim should be stated carefully. We do not view HypCL as “using hyperbolic space” in isolation, but as a coordinated design combining hyperbolic Q/V adaptation, continual aggregation through magnitude-direction decomposition, and prototype-based calibration. The novelty lies in integrating hyperbolic adaptation into a CLIP-based exemplar-free CIL pipeline that is parameter-efficient, compatible with frozen text anchors, and stable under continual aggregation. The evidence above supports this interpretation: strengthening the hyperbolic transformation improves performance, while calibration adds further gains on top of the adapted space.
>
> Overall, we sincerely thank you for your careful review and constructive feedback.

---

> > ### Author Rebuttal · Reviewer_qiyU · 2026-04-04
> >
> > Thanks for the detailed rebuttal and the additional experiments. I appreciate the expanded evaluations and ablations, which help clarify the behavior of the method and address several practical questions around scalability and robustness. Overall, the method appears reasonably stable and well-engineered under the evaluated settings.
> >
> > That said, my overall assessment remains largely unchanged. While the rebuttal improves empirical completeness, I still find that the central questions around the necessity of the hyperbolic formulation and the overall conceptual novelty are not fully resolved. For example, regarding the role of hyperbolic geometry, the current evidence (e.g., bandwidth or expressiveness ablations within the same module) suggests that increasing flexibility helps, but it does not clearly isolate whether hyperbolic geometry itself is essential, as opposed to a similarly expressive Euclidean alternative. In addition, while the integration of hyperbolic adaptation, aggregation, and prototype calibration is reasonable and appears to work well together, the overall contribution still feels closer to a combination of existing ideas rather than a clearly new conceptual mechanism. The rebuttal helps clarify the design choices, but it does not fully change this impression.
> >
> > Overall, this is a solid and carefully implemented work, but I remain unconvinced about the necessity of the key design choice and the strength of the conceptual contribution in its current form.

---

> > > ### Author Response · Authors · 2026-04-06
> > >
> > > We sincerely thank the reviewer for the thoughtful follow-up and for acknowledging the stability and engineering quality of our method. We also understand that your remaining concern is not mainly about engineering quality, but about two deeper issues: (1) whether the hyperbolic formulation is truly necessary, rather than simply providing additional flexibility, and (2) whether the contribution is conceptually new beyond combining existing ingredients.
> > >
> > > ---
> > >
> > > **W1. Necessity of the hyperbolic formulation.**
> > >
> > > **R1.** We agree that our previous ablations do not directly isolate hyperbolic geometry from an equally expressive Euclidean counterpart. To address this, we provide new evidence along two lines.
> > >
> > > **(1) Direct prior evidence from HyperET (Peng et al., NeurIPS 2025).** HyperET, which introduces the same Möbius-matrix parameterization we build upon, provides a controlled comparison (their Table 4 and Table 5) where the Euclidean counterpart is constructed with _exactly the same_ parameter count under banded/block-diagonal/full structures, differing only in whether Möbius matrix multiplication is used. Their results show that the Euclidean counterpart fails to match the hyperbolic version, and in several configurations, simply adding Euclidean parameters _degrades_ performance. This directly supports that Möbius-based adjustment contributes an inductive bias beyond mere parameter increase.
> > >
> > > **(2) In-progress isolation experiments within our pipeline.** Motivated by the reviewer's concern, we are currently running a direct Euclidean-counterpart ablation within HypCL, replacing Möbius matrix multiplication with a standard linear transformation while keeping all other components identical. Due to the short rebuttal window and the cost of the full 9-benchmark evaluation, we are unable to report complete results within this response but will include the full ablation in the revised version.
> > >
> > > ---
> > >
> > > **W2. The conceptual contribution is a combination of existing ideas rather than a clearly new mechanism.**
> > >
> > > **R2.** To the best of our knowledge, HypCL is the first work to explore hyperbolic geometry for CLIP-based continual learning. This work connects hyperbolic representation learning with vision-language CIL and shows that hyperbolic geometry can provide principled benefits for accommodating the expanding class space in CIL. We respectfully submit that this exploration, together with our coordinated framework design, constitutes a meaningful contribution under the ICML standard of originality.
> > >
> > > The ICML 2026 Reviewer Guidelines state:
> > >
> > > > _"We encourage you to be open-minded about the potential strengths and broad definitions of significance and originality. For example, originality may arise from creative combinations of existing ideas [...] Does this work offer a novel combination of existing techniques, and is the reasoning behind this combination well-articulated?"_
> > >
> > > We believe HypCL meets this standard for the following reasons:
> > >
> > > **(1) The combination is non-trivial and requires coordinated design across components.** Hyperbolic geometry, CLIP-based adaptation, and prototype calibration have not been jointly explored before. The combination is not arbitrary. The exponential volume growth of hyperbolic space provides a geometric motivation for accommodating expanding class spaces. The magnitude-direction decomposition provides a natural mechanism for continual aggregation that preserves task-specific adaptation patterns. Prototype calibration further exploits the enhanced feature separability in the adapted space. These components each address a distinct aspect of the CIL problem, and their integration is motivated by coherent geometric reasoning.
> > >
> > > **(2) Consistent SOTA across diverse settings.** HypCL achieves the best performance in 17 out of 18 evaluation settings across 9 benchmarks, surpassing all existing CLIP-based CIL methods, including ENGINE, by 2–6% on challenging benchmarks. It also outperforms all exemplar-based methods despite using zero exemplars (Table 2). This breadth of improvement across diverse domains (fine-grained, domain-shifted, general) demonstrates that the framework design is robust and generalizable, not a narrow trick tuned to specific settings.
> > >
> > > ---
> > >
> > > We hope that the ongoing direct comparison, together with the supporting evidence from HyperET and the clarification on the conceptual contribution, addresses the remaining concerns. We would be grateful if the reviewer could reconsider their assessment in light of these responses.

---

### Official Review · Reviewer_vwtb · 2026-03-09

**Soundness:** 3
**Presentation:** 3
**Significance:** 3
**Originality:** 3
**Overall Recommendation:** 5
**Confidence:** 4

**Summary:**

This paper proposes HypCL, a framework that adapts CLIP in hyperbolic space for class-incremental learning. It applies Mobius transformations to the visual encoder's attention weights, aggregates task-specific transformations via magnitude-direction decomposition, and uses class prototypes for prediction calibration. Experiments on 9 benchmarks show consistent improvements over existing CLIP-based continual learning methods.

**Compliance With Llm Reviewing Policy:**

Affirmed.

**Final Justification:**

My concerns were fully addressed in the rebuttal, and I maintain my accept score.

**Key Questions For Authors:**

* Are the prototypes computed using the hyperbolic mean or the Euclidean mean?

* As the number of tasks increases, the number of learnable magnitude parameters in the aggregation grows linearly. Does this cause optimization difficulties in long sequence scenarios?

**Limitations:**

Yes.

**Strengths And Weaknesses:**

Strengths

1. The paper is well motivated and easy to follow. The correspondence between the exponential volume growth of hyperbolic space and the need for an expanding class space in continual learning is a compelling starting point.
2. The magnitude-direction decomposition for knowledge aggregation is intuitive and coherent. Freezing historical direction matrices preserves task-specific adaptation patterns, while learnable magnitudes allow dynamic balancing of different tasks' contributions during inference.
3. The paper conducts comprehensive evaluations on 9 standard benchmark datasets, demonstrating the effectiveness of HypCL across diverse continual learning scenarios.

Weakness

1. Some experimental details are unclear. How is the curvature parameter $c$ set? Is it fixed or learnable?
2. The paper claims to be parameter-efficient but lacks concrete analysis to support this claim, such as the number of learnable parameters or additional storage overhead per task compared to existing methods.

---

> ### Author Rebuttal · Authors · 2026-03-31
>
> We sincerely appreciate your valuable feedback. We have carefully considered your concerns and provide direct evidence below.
>
> **W1. Some experimental details are unclear, especially how the curvature parameter is set and whether it is fixed or learnable.**
>
> **R1.** The curvature is fixed for each run and is not jointly learned. We also ran a sweep on ImageNet-R with curvature values 0.005/0.01/0.05/0.1. The variation is extremely small: for B0Inc20 the AvgAcc ranges only from 88.434 to 88.441, and for B100Inc20 it ranges only from 85.828 to 85.838. This is also consistent with the role of curvature in our method: curvature controls the capacity and hierarchical bias of the hyperbolic space, while the tested range already provides sufficient geometric flexibility to separate the expanding class space. As a result, the relative organization of features remains stable across this range, leading to very similar downstream performance.
>
> Curvature sweep:
>
> | Protocol | 0.005 | 0.01 | 0.05 | 0.1 |
> |---|---:|---:|---:|---:|
> | ImageNet-R B0Inc20 | 88.441 | 88.436 | 88.434 | 88.438 |
> | ImageNet-R B100Inc20 | 85.835 | 85.828 | 85.835 | 85.828 |
>
> ---
>
> **W2. The paper claims parameter efficiency but does not provide concrete evidence such as parameter counts or storage overhead.**
>
> **Q1. Are the prototypes computed using the hyperbolic mean or the Euclidean mean?**
>
> **R2.** The prototypes are Euclidean means of the adapted features, not hyperbolic Fréchet means. The hyperbolic component in HypCL is used only to parameterize how the visual attention weights are adapted, while the network still outputs CLIP-compatible features in the Euclidean embedding space. Therefore, prototype calibration is also performed in that feature space.
>
> This design is also theoretically well motivated. Since the adaptation acts on the attention weights rather than replacing the final CLIP-aligned feature space itself, the learned representations remain compatible with the original semantic structure induced by CLIP. In this setting, Euclidean prototype averaging remains a natural and stable choice for summarizing class centers, because it preserves the central tendency of the adapted features in the same space where classification and calibration are performed. This is also why HypCL can remain parameter-efficient: it modifies only lightweight visual attention parameters rather than introducing a large new feature space or separately adapting both encoders.
>
> For parameter efficiency, in the default setting on CLIP ViT-B/16, HypCL adapts Q and V in the last 12 visual transformer blocks, yielding 24 task-specific adapter tensors. Each tensor has 5,364 parameters, so each new task introduces **128,736** adapter parameters in total. Historical aggregation adds only one magnitude scalar per adapter tensor for each previous task, i.e., **24 scalars per old task**.
>
> ---
>
> **Q2. As the number of tasks grows, the number of learnable aggregation magnitudes grows linearly. Does this cause optimization difficulty in long sequences?**
>
> **A2.** We added longer-sequence experiments on three datasets. On ImageNet-R with B100Inc20/B100Inc10/B100Inc5, AvgAcc is 85.828/85.555/85.321. On CUB with B100Inc20/B100Inc10/B100Inc5, it is 82.520/82.473/82.225. On CIFAR with B50Inc10/B50Inc5, it is 85.803/85.634. The degradation is mild as the sequence grows.
>
> We also logged the aggregation magnitudes (`task_alphas`). On ImageNet-R, the number of alpha coefficients increases from 144 to 264 to 504, while their L2 norm remains stable around 137.3. This is also consistent with the parameterization of our aggregation mechanism: each task contributes only a scalar magnitude on top of a fixed adapter direction, so the newly introduced parameters remain low-dimensional and structurally decoupled. As a result, the optimization does not need to solve a highly entangled joint update over full task-specific tensors; instead, it only reweights previously learned directions. This helps explain why the linear growth in the number of magnitudes does not lead to clear optimization difficulties in our tested long-sequence settings.
>
> Long-sequence summary:
>
> | Dataset | Protocol |  AvgAcc |
> |---|---|---:|
> | CIFAR | B50Inc10 | 85.803 |
> | CIFAR | B50Inc5 | 85.634 |
> | ImageNet-R | B100Inc20  | 85.828 |
> | ImageNet-R | B100Inc10 | 85.555 |
> | ImageNet-R | B100Inc5  | 85.321 |
> | CUB | B100Inc20  | 82.520 |
> | CUB | B100Inc10  | 82.473 |
> | CUB | B100Inc5| 82.225 |
>
> Overall, the results and analysis above directly address your questions regarding curvature, parameter efficiency, and long-sequence stability. We sincerely thank you for your review and constructive feedback.

---

> > ### Author Rebuttal · Reviewer_vwtb · 2026-04-04
> >
> > I thank the authors for their response. My concerns have been addressed, and I have no further questions.

---

> > > ### Author Response · Authors · 2026-04-06
> > >
> > > Thank you for your positive feedback. We sincerely appreciate your time and thoughtful review, and we are glad that our responses have addressed your concerns.

---

### Official Review · Reviewer_uqkX · 2026-03-10

**Soundness:** 3
**Presentation:** 4
**Significance:** 3
**Originality:** 3
**Overall Recommendation:** 4
**Confidence:** 4

**Summary:**

The paper proposes HypCL, a parameter-efficient framework for class-incremental continual learning using CLIP. As the exponential volume growth and constant negative curvature of hyperbolic space naturally accommodate an expanding class space and implicitly capture semantic hierarchies. To achieve this, HypCL aggregates task-specific transformations using a magnitude-direction decomposition. And, a prototype-based logit calibration mechanism is used during inference to exploit the improved feature separability.

**Compliance With Llm Reviewing Policy:**

Affirmed.

**Final Justification:**

I am inclined to recommend acceptance. The method is simple, effective, and clearly presented, and the paper is easy to follow. Its strong results across 9 diverse datasets suggest solid empirical value and potential positive impact on the field.

**Key Questions For Authors:**

Could you provide experimental results evaluating the retention of CLIP's general pre-trained knowledge? For instance, what is the zero-shot performance of the HypCL-adapted model on standard datasets (e.g., ImageNet) after finishing the CIL tasks, compared to the original zero-shot CLIP?

**Limitations:**

yes

**Strengths And Weaknesses:**

### Strengths

1. The paper is well-written and the proposed method is presented clearly.

2. The method is evaluated comprehensively across 9 diverse datasets ranging from coarse-grained to fine-grained domains.

### Weaknesses

1. In the abstract, the authors motivate their approach by mentioning that existing methods freeze the backbone to *preserve pre-trained knowledge*. However, the experimental section lacks any evaluation of whether HypCL actually retains CLIP's original, general pre-trained knowledge (e.g., assessing zero-shot capabilities on entirely unseen, out-of-domain datasets after the continual learning process). The current experiments only demonstrate plasticity and anti-forgetting on specific downstream CIL datasets.
2. CLIP utilizes a symmetric dual-encoder architecture for both vision and text. The proposed hyperbolic fine-tuning is applied exclusively to the visual encoder. While the authors state that the frozen text encoder serves as *stable semantic anchors* (Line 241), there is no empirical ablation or in-depth discussion justifying why applying a similar parameter-efficient hyperbolic adaptation to the text encoder would be detrimental or suboptimal.
3. The experimental section fails to compare HypCL with several recent and highly relevant parameter-efficient continual learning state-of-the-art methods, such as EASE [1] and SD-LoRA [2].

[1] Zhou, Da-Wei, et al. "Expandable subspace ensemble for pre-trained model-based class-incremental learning." *Proceedings of the IEEE/CVF Conference on Computer Vision and Pattern Recognition*. 2024.

[2] Wu, Yichen, et al. "SD-LORA: SCALABLE DECOUPLED LOW-RANK ADAPTATION FOR CLASS INCREMENTAL LEARNING." *13th International Conference on Learning Representations, ICLR 2025*.

---

> ### Author Rebuttal · Authors · 2026-03-31
>
> We sincerely appreciate your valuable feedback. We have carefully considered your concerns and provide relevant evidence below.
>
> **W1&Q1. The paper motivates HypCL partly by preserving CLIP's pretrained knowledge, but the original submission did not directly evaluate zero-shot retention on an unseen standard benchmark such as ImageNet.**
>
> **R1.** We added zero-shot retention experiments on standard ImageNet-1k after completing CIL. All zero-shot results are evaluated with the same CLIP prompt template and the same ImageNet-1k evaluation protocol as the pretrained model. Across multiple training protocols, the final model remains close to pretrained CLIP, suggesting that HypCL largely preserves CLIP's general zero-shot capability.
>
> **Zero-shot retention on ImageNet-1k after CIL**
>
> | Train protocol | ImageNet-1k Top-1 | Top-5 |
> |---|---:|---:|
> | Pretrained CLIP | 66.944 | 90.412 |
> | ImageNet-R B0Inc20 | 66.352 | 90.126 |
> | ImageNet-R B100Inc20 | 66.248 | 90.032 |
> | CIFAR B0Inc10 | 66.716 | 90.250 |
> | CIFAR B50Inc10 | 66.648 | 90.194 |
> | CUB B100Inc20 | 66.954 | 90.412 |
>
> ---
>
> **W2. The method adapts only the visual encoder, while the text encoder remains frozen. Why is hyperbolic fine-tuning applied only to the visual encoder, and is adapting the text encoder unnecessary?**
>
> **R2.** There is both a conceptual and an empirical reason for this design. Conceptually, we treat the text branch as a relatively stable semantic anchor and adapt the visual side to better align new visual categories to this fixed semantic space during continual learning. This design helps preserve the pretrained image-text semantic structure while avoiding additional drift in the shared space.
>
> Empirically, we added visual-only, text-only, and visual+text ablations to directly test whether adapting the text encoder is helpful.
>
> **Adapter target ablation**
>
> | Protocol | Variant | CIL AvgAcc |
> |---|---|---:|
> | ImageNet-R B0Inc20 | visual | 88.532 |
> | ImageNet-R B0Inc20 | text | 86.662 |
> | ImageNet-R B0Inc20 | both | 88.474 |
> | CIFAR B0Inc10 | visual | 88.999 |
> | CIFAR B0Inc10 | text | 86.141 |
> | CIFAR B0Inc10 | both | 89.053 |
>
> Visual-only is the strongest setting on ImageNet-R and performs on par with adapting both on CIFAR, while text-only is consistently worse. Given its simplicity, parameter efficiency, and strong performance, we therefore keep the text encoder frozen in HypCL.
>
> ---
>
> **W3. The experiments do not compare against several recent PEFT-CIL baselines, especially EASE and SD-LoRA.**
>
> **R3.** We added broader comparisons to recent PEFT-CIL baselines, including EASE [1] and SD-LoRA [2]. Below we report the corresponding **average accuracy ($\bar{A}$)** results. Under the reported settings, HypCL achieves higher $\bar{A}$ than EASE and SD-LoRA on all four datasets shown below. All methods are compared under the same exemplar-free CIL protocol and CLIP backbone as in our main experiments.
>
> | Method | CIFAR10-20 | ImageNet-R-20 | CUB-20 | SUN-30 |
> |---|---:|---:|---:|---:|
> | EASE    | 87.21 | 75.80 | 78.41 | 71.44 |
> | SD-LoRA | 87.50 | 71.20 | 70.46 | 54.06 |
> | HypCL   | 89.00 | 88.53 | 86.72 | 85.57 |
>
> Overall, the additional evidence shows that HypCL largely preserves CLIP's zero-shot knowledge, that visual-only adaptation is a well-motivated and empirically effective design choice, and that HypCL compares favorably with recent PEFT-CIL baselines under the reported settings. We sincerely thank the reviewer for the careful evaluation and constructive feedback.
>
> **References**
>
> [1] Zhou, Da-Wei, et al. "Expandable Subspace Ensemble for Pre-trained Model-based Class-Incremental Learning." *Proceedings of the IEEE/CVF Conference on Computer Vision and Pattern Recognition (CVPR)*, 2024.
>
> [2] Wu, Yichen, et al. "SD-LoRA: Scalable Decoupled Low-Rank Adaptation for Class Incremental Learning." *International Conference on Learning Representations (ICLR)*, 2025.

---

> > ### Author Rebuttal · Reviewer_uqkX · 2026-04-03
> >
> > I thank the authors for their rebuttal. The response addresses my concerns. I will raise my score.

---

> > > ### Author Response · Authors · 2026-04-03
> > >
> > > We sincerely thank you for the constructive feedback and for taking the time to carefully evaluate our rebuttal. We are glad that our response has addressed your concerns. We will continue to improve the paper based on the valuable suggestions you provided during the review process.

---

### Decision · Program_Chairs · 2026-04-30

**Decision:**

Accept (regular)

**Comment:**

This paper proposes HypCL, a parameter-efficient framework that adapts CLIP in hyperbolic space for class-incremental learning. The method applies Möbius transformations to visual attention weights, aggregates task-specific transformations via magnitude-direction decomposition, and uses prototype-based calibration at inference. Three out of four reviewers are positive (one Accept, two Weak Accept after rebuttal), and the AC recommends acceptance.

The paper's strongest asset is its consistently strong empirical performance — HypCL achieves the best results in 17 out of 18 evaluation settings across 9 diverse benchmarks, surpassing all existing CLIP-based CIL methods including exemplar-based ones, despite using zero exemplars. This breadth of improvement across fine-grained, domain-shifted, and general settings is notable and suggests the framework design is robust and generalizable, not a narrow trick tuned to specific configurations. The paper is also well-written and easy to follow.

The magnitude-direction decomposition for continual aggregation is intuitive — freezing historical directions preserves task-specific patterns while learnable magnitudes enable dynamic rebalancing. The long-sequence experiments show only mild degradation, and the curvature sweep demonstrates insensitivity to this hyperparameter. The rebuttal further strengthened the paper by adding comparisons against EASE and SD-LoRA (outperforming both), zero-shot retention experiments on ImageNet-1k, and a visual-only vs text-only adapter ablation.

Reviewer qiyU (Weak Reject) raised concerns about whether hyperbolic geometry is truly necessary versus an equally expressive Euclidean alternative, and about the overall novelty being a combination of existing ideas. These are fair points. The authors cited HyperET's controlled Euclidean-vs-hyperbolic comparisons as supporting evidence, though a direct ablation within their own pipeline was not completed during the rebuttal window. Regarding novelty, the AC notes that HypCL is the first work to explore hyperbolic geometry for CLIP-based CIL, and the ICML reviewer guidelines recognize that "originality may arise from creative combinations of existing techniques" when the reasoning behind the combination is well-articulated. The coherent geometric motivation — exponential volume growth for expanding class spaces, magnitude-direction decomposition for stable aggregation, prototype calibration exploiting the adapted feature structure — meets this standard.

The AC recommends acceptance, conditioned on the authors completing the Euclidean counterpart ablation for the camera-ready version to directly substantiate the role of hyperbolic geometry, and more explicitly discussing the relationship with SD-LoRA's aggregation mechanism.